# MCM-aware Twin-least-square GAN for Hyperspectral Anomaly Detection

## Abstract

Hyperspectral anomaly detection under high-dimensional data and interference of deteriorated bands without any prior information has been challenging and attracted close attention in the exploration of the unknown in real scenarios. However, some emerging methods based on generative adversarial network (GAN) suffer from the problems of gradient vanishing and training instability with struggling to strike a balance between performance and training sample limitations. In this work, aiming to remedy the drawbacks of existing methods, we present a novel multi-scale covariance map (MCM)-aware twin-least-square GAN (MTGAN). Instead of the widely used single-scale Gaussian hypothesis background estimation, in MTGAN, we introduce the MCM-aware strategy to construct multi-scale priors with precise second-order statistics, thereby implicitly bridging the spatial and spectral information. Thus, we reliably and adaptively represent the prior of HSI to change the priors-lack situation. Moreover, we impose the twin-least-square loss on GAN, which helps improve the generative ability and training stability in feature and image domains, overcoming the gradient vanishing problem. Finally, the network enforced with a new anomaly rejection loss establishes a pure and discriminative background estimation. Experiments demonstrate that the average detection accuracy of MTGAN reaches 0.99809, which is superior to the state-of-the-art algorithms.

## 1 Introduction

Hyperspectral image (HSI) appears as a three-dimensional (3D) data cube, two dimensions of which show the spatial information of materials, and the other reveals hundreds of contiguous bands to perceive each scene (Yokoya et al., 2012). Among a wealth of HSIs interpretation techniques in practical situations, anomaly detection has many potential applications in video surveillance, activity recognition, and scene understanding, etc (Lanaras et al., 2015; Eyal et al., 2019; Tu et al., 2020). However, due to the insufficient prior information, inaccurate labels, complex scenes, and unbalanced samples, it is high-cost and sometimes infeasible to accurately detect different types of anomalies in HSI. Consequently, hyperspectral anomaly detection without any priors is a challenging task and is of great importance.

Deep learning-based methods have powerful and unique advantages in modeling and characterizing complex data (Stanislaw et al., 2020). A lot of research has appeared in the field of anomaly detection, which can be roughly divided into three categories: supervised, semi-supervised, and unsupervised. However, due to the difficulty of annotation and collection of label training, supervised methods are rarely applied (Grnitz et al., 2013; Raghavendra & Sanjay, 2019). Semi-supervised work aims to break the dilemma between the number of samples and detection performance, but it still requires pure background training samples (Blanchar et al., 2010; Wu & Prasad, 2018). On the one hand, unsupervised learning based hyperspectral anomaly detection has become a new trend (Schlegl et al., 2017; Zhang et al., 2019). On the other hand, the detection performance is limited due to the lack of prior knowledge. Therefore, we propose an MCM-aware strategy to adaptively obtain reliable and stable pseudo-labeled prior information to alleviate these problems.

Concretely, motivated by the observations mentioned above, we estimate the priors and model the background with multi-scale covariance matrices as the necessary preparation fed into the MTGAN model, which generates discriminative representations with second-order statistics in covariance

pooling and is conducive to exploiting the intrinsic spatial-spectral information of HSI. The progress of MCM-aware priors construction strategy is illustrated in Figure 1.

Furthermore, though GAN performs well in anomaly detection tasks according to the literature, the real objective of GAN is supposed to capture more separable latent features between background and anomalies instead of minimizing the pixel-wise reconstruction error (Gong et al., 2020). The gradient vanishing problem, which is partly caused by the hypothesize that the discriminator as a classifier with the sigmoid cross-entropy loss function in regular GANs, is not conducive to the generation of background and discrimination of anomalies.

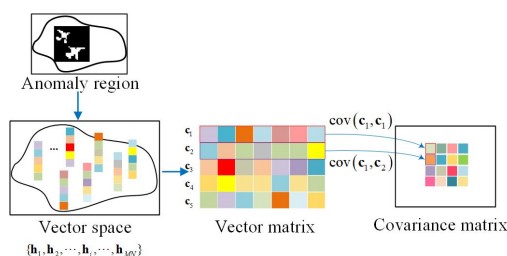

Figure 1: The progress of MCM-aware priors construction strategy.

Hence, to facilitate the training stability and alleviate the gradient-vanishing problem, we present twin-least-square loss to perform background modeling in feature and image domains. Accordingly, we can solve the problem of gradient vanishing and enhance the representation directly aiming at the reconstruction of each pixel.

In light of the difficulties of the separability between the anomaly and background, we also impose an anomaly rejection loss to avoid anomalies contamination in background estimation. In this way, the network can reconstruct resembled background dictionaries, but dramatically changed anomalies, thereby increasing the degree of difference between them and endow better detection accuracy.

To verify the effectiveness of the proposed method, we implement evaluations on five public HSI data sets. In MTGAN, the average AUC scores of $(P_d, P_f)$ and $(P_f, \tau)$ are 0.99809 and 0.00518, respectively, which outperform previous state-of-the-art methods. To summary, our contributions are mainly three-fold:

- To solve the problem of insufficient samples that previous methods suffer from, we propose an MCM-aware strategy to reliably and adaptively generate prior dictionaries. In specific, we calculate a series of multi-scale covariance matrices, taking advantage of the second-order statistics to naturally model the distribution with integrated spectral and spatial information.

- The twin-least-square loss is introduced into both the feature and image domains to overcome the gradient vanishing problem. Meanwhile, the generative ability and training stability can be improved, which can fit the characteristics of high-dimension and complexity of HSI data.

- To further reduce the false alarm rate, we design a novel anomaly rejection loss to enlarge the distribution diversity between background regions and anomalies, aiming to distinguish between background and anomalies. Experimental results illustrate that the AUC score of $(P_f, \tau)$ in MTGAN is one order of magnitude lower than other state-of-the-art methods.

## 2 RELATED WORK

For traditional methods, the RX method assumes that each spectral channel is Gaussian-distributed, and the pixel is L-dimensional multi-variate Gaussian distributed (Guo et al., 2014; Luo et al., 2019; Ahmed et al., 2020). As a non-RX based methods, the ADLR method obtains abundance vectors by spectral decomposition and constructs a dictionary based on the mean value clustering of abundance vectors (Qu et al., 2018). The PAB-DC model imposed with low-rank and sparse constraints considers the homogeneity of background and the sparsity of anomalies to construct the dictionaries (Huyan et al., 2019). The emerging typical algorithm AED removes the background mainly by attribute filtering and difference operation. Additionally, the LSDM-MoG method combines the mixed noise models and low-rank background to characterize complex distributions more accurately (Li et al., 2020). However, these conventional methods are based on single-scale Gaussian assumption and cannot represent complex and high-dimensional data sets well, leading to the exploration of deep learning-based methods (Ben et al., 2014).

In deep auto-encoding Gaussian mixture model (DAGMM) (Zong et al., 2018), an autoencoder (AE) is introduced to the model to generate a low-dimensional representation and reconstruction error for each input data point as the input of the Gaussian Mixture Model (GMM). GAN has attracted a lot of attention for providing a generative model to minimize the distance between the training data distribution and the generative model samples without explicitly defining a parametric function (Goodfellow et al., 2014; Yuan et al., 2019; Gu et al., 2020). A novel single-objective generative adversarial active learning (SO-GAAL) method for outlier detection considers the anomaly detection as a binary-classification issue by sampling potential outliers from a uniform reference distribution (Liu et al., 2019). Nevertheless, these deep learning-based methods cannot achieve a balance between good performance and limited prior information. What's more, the network structure of these methods is not specially designed for hyperspectral anomaly detection. Therefore, we propose MT-GAN concerning hyperspectral anomaly detection for the first time, to approximate the performance of the supervised methods while releasing the limitation of training samples.

## 3 PROBLEM STATEMENT AND FRAMEWORK

In this work, we elaborate on MTGAN for hyperspectral anomaly detection, as shown in Figure 2. The three key components of the framework include: 1) the MCM module for background dictionary construction; 2) the twin-least-square GAN module for background reconstruction; 3) the anomaly rejection loss added joint learning. The modules are cascaded together for hyperspectral anomaly detection.

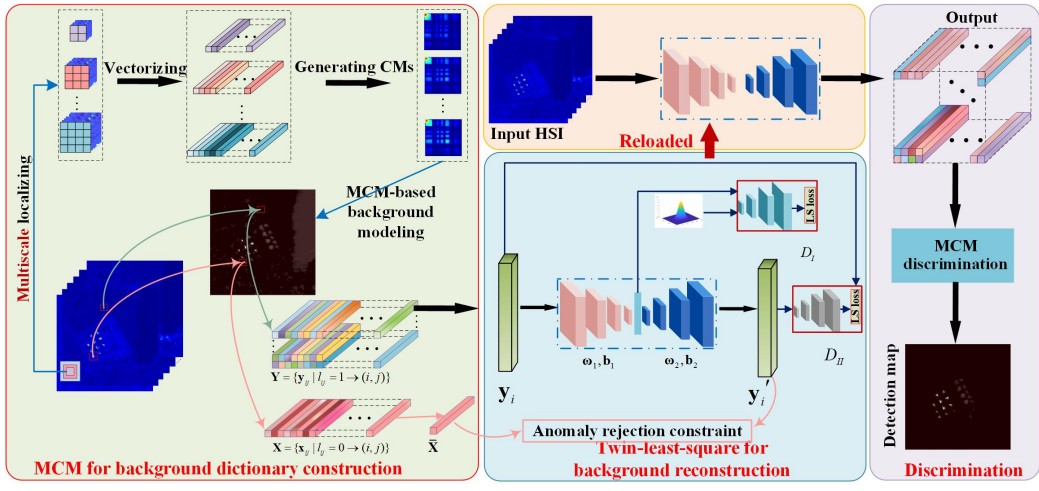

Figure 2: Overview of our proposed MTGAN framework.

### 3.1 CONSTRUCTING THE OVERALL MODEL

We denote the HSI as $\mathbf{H} \in \mathbb{R}^{h \times w \times d}$, where $d$ is the number of the spectral bands. $h$ and $w$ represent the spatial size of the data. For convenience, as the input of the network, we transform the 3-D cube $\mathbf{H}$ into a 2-D matrix $\mathbf{H} = \{\mathbf{h}_i\}_{i=1}^{n} \in \mathbb{R}^{d \times n}$, where each column of $\mathbf{H}$ is a spectral pixel vector in the HSI and $n = h \times w$ is the number of the pixels. The HSI data matrix is decomposed into two components: background and anomaly. We denote background and anomaly as $\mathbf{Y} = [\mathbf{y}_1, \mathbf{y}_2, ..., \mathbf{y}_{n_B}]$, $\mathbf{X} = [\mathbf{x}_1, \mathbf{x}_2, ..., \mathbf{x}_{n_A}]$, and $n_A + n_B = n$, respectively, where $\mathbf{y}_i$ and $\mathbf{x}_i$ represent the $i$th vectors.

Based on the defined background and anomaly dictionaries, we formulate the anomaly detection method as

$$L\left(\mathbf{Y}, \mathbf{X}\right) = L_{TLS}\left(\mathbf{Y}\right) + L_{auto}\left(\mathbf{Y}, \hat{\mathbf{Y}}\right) + L_{enlarge}\left(\hat{\mathbf{Y}}, \mathbf{X}\right)$$
$$= L_{LS_1}\left(\mathbf{Z}\right) + L_{LS_2}\left(\mathbf{Y}, \hat{\mathbf{Y}}\right) + \left\|\mathbf{Y} - \hat{\mathbf{Y}}\right\| - \alpha \left\|\hat{\mathbf{Y}} - \mathbf{X}\right\|, \quad (1)$$

$$s.t. \begin{cases} \mathbf{Z} = Enc\left(\mathbf{Y}; \arg(G)\right) \\ \hat{\mathbf{Y}} = Dec\left(\mathbf{Z}; \arg(D)\right) \\ \alpha \sim N\left(0, I\right) \end{cases},$$

where $\left\| \mathbf{Y} - \hat{\mathbf{Y}} \right\|$ denotes the reconstruction error of the basic AE network. The twin-least-square loss $L_{TLS}$ added for the two discriminators are denoted by $L_{LS_1}$ and $L_{LS_2}$, respectively, which make up the whole twin-least-square loss $L_{TLS}$. $L_{auto}$ and $L_{enlarge}$ represent the spectral reconstruction loss and separability loss between background-anomaly, respectively. $Enc$ and $Dec$ represent the encoder and decoder, respectively. $\mathbf{Z}$ is the output of the encoder. And the encoding and decoding process can be demonstrated as

$$\hat{\mathbf{Y}} = \sigma\left(\mathbf{W}\mathbf{W}^T\mathbf{Y} + \mathbf{B}\right), \tag{2}$$

where $\sigma\left(\cdot\right)$ represents the activation function. $\hat{\mathbf{Y}}$ is the output of the network, and $\mathbf{W}$ is the weight of the encoder. $\mathbf{B}$ is the bias of the whole network.

## 3.2 MCM-aware Prior for Background Construction

Inspired by estimating the Mahalanobis distance between the test pixels and the constructed pixels on one scale, we generate pseudo priors for GAN training through multi-scale covariance maps construction. Thus, we can meet the requirement of sufficient prior information and take advantage of the spatial and spectral information of HSI. The whole generation progress by pseudo-labeling can be expressed as

$$(\mathbf{Y}, \mathbf{X}) = f_{MCM}\left(\mathbf{H}\right), \tag{3}$$

where $f_{MCM}\left(\cdot\right)$ represents the nolinear learning process of MCM strategy. $\mathbf{Y}$ and $\mathbf{X}$ denote the background and anomaly dictionaries, respectively.

### 3.2.1 Multi-scale Localizing

For each central pixel, we try to realize multi-scale localizing first based on the Euclidean distance with a classical classifier, i.e., K nearest neighbors (KNN), to obtain the local pixel cubes at different scales. Then, we generate a series of gradually increasing cubes of different scales. For each of the cubes, we transfer it to a vector. After that, the covariance matrix is calculated between the vectors.

### 3.2.2 Generating Co-variance Maps

For the central pixel $\mathbf{h}_i$, taking the scale $R \times R$ as an example, the covariance map of $\mathbf{h}_i$ on the fixed one scale is extracted as

$$\mathbf{C}_k = \frac{1}{R^2 - 1} \sum_{i=1}^{R^2} \left(\mathbf{h}_i - \mu\right)\left(\mathbf{h}_i - \mu\right)^T \in \mathbb{R}^{L \times L}, \tag{4}$$

where $\mu$ represents the mean of the set of input HSI vectors $\{\mathbf{h}_i | i = 1, 2, \cdots, R\}$. $\{\mathbf{h}_i | i = 2, \cdots, R^2\}$ represents the corresponding adjacent pixels in a window of $R \times R$ pixels. In addition, M scales of $R_k$, i.e. $k = 1, \cdots, M$, are taken into account. The covariance maps of other scales are denoted by $\mathbf{C}_k$, $k = 1, \cdots, M$, which make up the co-variance pool to construct the background.

## 3.3 Cascaded Architecture with Twin-least-square loss

### 3.3.1 Stability Branch

As mentioned above, as the improved version of the original GAN, we present the architecture based on twin-least-square loss to solve the gradient vanishing problem and enhance the stability. As shown in Figure 2, the MTGAN network consists of two discriminators and a generator. Instead of the cross-entropy in GAN, we impose twin-least-square loss on the two discriminators. The twin-least-square loss can be expressed as

$$L_{LS_1} = \frac{1}{2} E_{\mathbf{y} \sim p_{data}(\mathbf{y})}[(D_F(\mathbf{y}) - 1)^2] + \frac{1}{2} E_{\mathbf{z} \sim p_{\mathbf{z}}(\mathbf{z})}[(D_F(G(\mathbf{z})) + 1)^2], \tag{5}$$

$$L_{LS_2} = \frac{1}{2} E_{\mathbf{y} \sim p_{data}(\mathbf{y})}[(D_R(\mathbf{y}) - 1)^2] + \frac{1}{2} E_{\hat{\mathbf{y}} \sim p_{\hat{\mathbf{y}}}(\hat{\mathbf{y}})}[(D_R(G(\hat{\mathbf{y}})) + 1)^2]. \tag{6}$$

From the binary classification point of view, we introduce the twin-least-square loss to move false samples to the decision boundary of being anomaly or background and punish those being far from the decision boundary, even if on the right side. Thus, we perform adversarial training on the two least-square-loss imposed discriminators $D_R$ and $D_F$ against the generator, aiming to overcome gradient vanishing and stably match the distribution of the decoded vectors $\hat{\mathbf{y}}$ and the known input data distribution of $\mathbf{y}$.

### 3.3.2 SEPARABILITY BRANCH

As mentioned earlier, we impose the spectral reconstruction loss of AE using mean squared error (MSE) to minimize the deviation between the decoded images and the original input image:

$$\begin{aligned} L_{auto}\left(\mathbf{Y}, \hat{\mathbf{Y}}\right) &= \left\|\mathbf{Y} - \hat{\mathbf{Y}}\right\| \\ &= \left\|\mathbf{Y} - \sigma\left(\mathbf{W}\mathbf{W}^T\mathbf{Y} + \mathbf{B}\right)\right\|. \end{aligned} \tag{7}$$

To ensure that the learning samples come entirely from the background, we introduce the following distance function based on $L_{auto}$. The second item of which is expected to be as large as possible:

$$\begin{aligned} L_{anorm}\left(\mathbf{Y}, \hat{\mathbf{Y}}\right) &= L_{auto}\left(\mathbf{Y}, \hat{\mathbf{Y}}\right) + L_{enlarge}\left(\hat{\mathbf{Y}}, \mathbf{X}\right) \\ &= \left\|\mathbf{Y} - \hat{\mathbf{Y}}\right\| - \alpha\left\|\hat{\mathbf{Y}} - \mathbf{X}\right\| \\ &= \left\|\mathbf{Y} - \sigma\left(\mathbf{W}\mathbf{W}^T\mathbf{Y} + \mathbf{B}\right)\right\| - \alpha\left\|\sigma\left(\mathbf{W}\mathbf{W}^T\mathbf{Y} + \mathbf{B}\right) - \mathbf{X}\right\|. \end{aligned} \tag{8}$$

Let $\mathbf{x}_i$ and $\mathbf{y}_i$ represent the $i$th component in $\mathbf{X}$ and $\mathbf{Y}$, respectively. $\bar{\mathbf{x}}$ denotes the mean of all the $\mathbf{x}_i$ in $\mathbf{X}$. When the distance between the reconstructed spectrum vector $\hat{\mathbf{y}}_i$ and the average spectrum vector $\bar{\mathbf{x}}$ is small, $\mathbf{y}_i$ is suspected to be the anomaly. Then the suppression coefficient $\alpha$ of the function aims to reduce it rapidly for adjustment. When the distance between $\bar{\mathbf{x}}$ and $\mathbf{y}_i$ is large, from a statistical point of view, $\mathbf{y}_i$ is the background dictionary to be estimated. Then the value of the suppression coefficient $\alpha$ is approximate to 1.

### 3.3.3 JOINT TRAINING AND DISCRIMINATION

The twin-least-square loss, spectral reconstruction loss, and anomaly rejection loss are jointly learned with the weighting coefficient of 1 by doing alternative updates of each component as follows. Subsequently, we obtain the detection maps by Gaussian statistics-based discrimination.

- Minimize $L_{LS_1}$ by updating parameters of $D_F$.
- Minimize $L_{LS_2}$ by updating parameters of $D_R$.
- Minimize $L_{anorm}$ by updating parameters of $En$ and the decoder $De$.

## 4 EXPERIMENT

### 4.1 EXPERIMENTAL SETUP

**Data Sets.** HYDICE describes urban scenes in the United States that were captured by the hyperspectral digital image acquisition experiment (HYDICE) airborne sensors above the city. A sub-image with a spatial size of $80 \times 100$ is cut out from the entire $307 \times 307$ original image with 162 spectral channels ranging from 400 nm to 2500 nm. Airport-Beach-Urban (ABU) was captured by the Airborne visible/infrared imaging spectrometer (AVIRIS) sensors. Among airport, beach, and urban scenes, we conduct experiment on two of the urban scenes. The sample images in ABU contain $100 \times 100$ or $150 \times 150$ pixels and so as the corresponding reference maps. EI Segundo is captured by AVIRIS sensor. The scene contains the area of the refinery, several residential areas, parks, and school areas. It has $250 \times 300$ pixels in the spatial domain and 224 bands in spectral,

ranging from 366 to 2496 nm. Grand Island is also acquired by the AVIRIS sensor at the location of Grand Island on the Gulf Coast, which contains $300 \times 480$ pixels and 224 spectral channels with a wavelength range of 366 to 2496 nm. The pseudo-color images of each data set and their corresponding ground truth maps are shown in Figure 4, respectively.

**Hyper-parameters.** In our experiment, we set the number of training iterations as 5000. The learning rate is set to 0.0001. There are three layers for the encoder, decoder, and discriminators networks, respectively. The number of the extracted feature is set to 20. With back propagation, we optimize all parameters by the Adam optimization algorithm.

## 4.2 EVALUATION RESULTS

For quantitative comparison, we can observe from Tables 1 and 2 that for all data sets, the results of the MTGAN method and comparison methods are close to ideal values.

| Method | HYDICE | urban1 | urban2 | Grand Island | EI Segundo | Average |
|--------|--------|--------|--------|--------------|------------|---------|
| RX | 0.97637 | 0.99463 | 0.98870 | 0.99990 | 0.97826 | 0.98757 |
| LSDM-MoG | 0.95643 | 0.99321 | 0.97504 | 0.99989 | 0.96798 | 0.97851 |
| ADLR | 0.99471 | 0.98774 | 0.99223 | **0.99993** | 0.99145 | 0.99321 |
| PAB-DC | 0.99760 | 0.98327 | 0.54635 | 0.98768 | 0.75911 | 0.85480 |
| AAE | 0.92185 | 0.96806 | 0.99284 | 0.99940 | 0.99260 | 0.97495 |
| MTGAN | **0.99945** | **0.99816** | **0.99312** | 0.99991 | **0.99982** | **0.99809** |

Table 1: Evaluation AUC score of $(P_d, P_f)$ obtained by MTGAN and compared methods.

| Method | HYDICE | urban1 | urban2 | Grand Island | EI Segundo | Average |
|--------|--------|--------|--------|--------------|------------|---------|
| RX | 0.03798 | 0.01351 | 0.01140 | 0.00738 | 0.00952 | 0.01596 |
| LSDM-MoG | 0.18799 | 0.04555 | 0.01338 | 0.02848 | 0.14127 | 0.08333 |
| ADLR | **0.00316** | 0.08011 | 0.07451 | 0.00079 | 0.09622 | 0.05096 |
| PAB-DC | 0.10525 | 0.16974 | 0.32350 | 0.06799 | 0.10506 | 0.15431 |
| AAE | 0.00962 | 0.04229 | 0.06192 | 0.04818 | 0.04355 | 0.04111 |
| MTGAN | 0.01541 | **0.00200** | **0.00145** | **0.00069** | **0.00218** | **0.00518** |

Table 2: Evaluation AUC score of $(P_f, \tau)$ obtained by MTGAN and compared methods.

Specifically, for the HYDICE data set, though the $(P_f, \tau)$ obtained by MTGAN does not reach the order of magnitude of three decimal places, the AUC score of $(P_d, P_f)$ is 0.99945 which outperforms other methods, demonstrating that the MTGAN can maintain the detection ability well with extremely low missed detection. Compared with RX and PAB-DC methods, for the ABU-urban1 data set, MTGAN can detect more anomalies, showing better performance. According to the results in Table 1, the AUC score of $(P_f, \tau)$ of MTGAN is smaller than other comparison methods for the ABU-urban-1 data set, indicating that the MTGAN can inhibit background for the scenarios better. For Grand Island, the AUC score of $(P_d, P_f)$ obtained by MTGAN is 0.99991, which is much higher than most typical methods but less than the ADLR algorithm of 0.99993. However, the AUC score of $(P_f, \tau)$ of MTGAN is 0.00069, which is better than the ADLR of 0.00079. The detection of the ABU-urban2 and EI Segundo data sets also performs well, which indicates that MTGAN has compromised the AUC scores of $(P_d, P_f)$ and $(P_f, \tau)$. Generally speaking, the comparison methods may achieve good results for some specific data sets, but MTGAN can achieve promising detection results both in the AUC scores of $(P_d, P_f)$ and $(P_f, \tau)$ for all the data sets. From the ROC curves shown in Figure 3, we can conclude that the MTGAN obtains higher AUC score of $(P_d, P_f)$ with identical AUC score of $(P_f, \tau)$, illustrating a superior performance to other methods.

## 4.3 PARAMETER SENSITIVITY ANALYSIS

There are mainly two parameters to be analyzed: the size of the window and the scales of the MCMs. Due to the pixel-level processing characteristics of the MTGAN and its sensitivity to the size of the

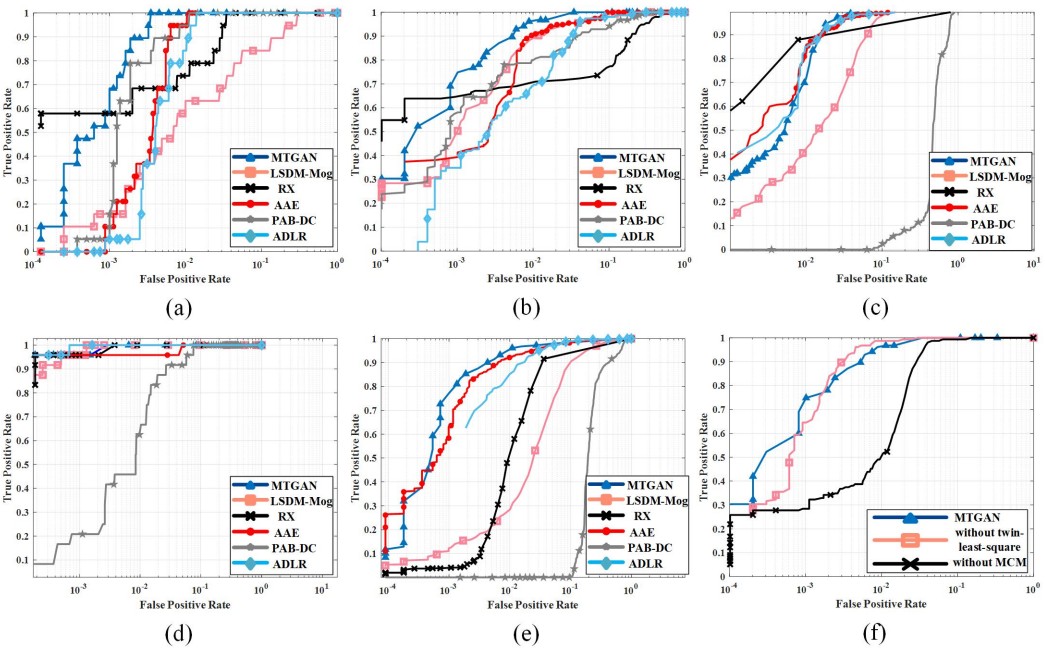

Figure 3: The ROC curves of different methods for (a) HYDICE, (b) ABU-urban1, (c) ABU-urban2, (d) Grand Island, (e) EI Segundo, (f) models in MTGAN without MCM or twin-least-square loss.

windows, we fixed the scales of the MCMs as 5. With the size of the windows varying from 30 to 50, the values differ from each other, and the optimal results are achieved for different data sets. For the HYDICE data set, it achieves the highest AUC score of $(P_d, P_f)$ when the size is 25. And the value is 15 for the ABU-urban-2 data set. As for the Grand Island and EI Segundo data sets, the optimal size values are 20 and 24, respectively.

Similarly, when the scales of the MCM are changed from 2 to 10 at 2 steps, the results of the AUC score of $(P_d, P_f)$ illustrate different performance. When we set the scales to 6 for the HYDICE data set, the performance of the experimental data set can achieve the best. For the rest of the data sets, we set the scales of the MCM as 5 to make a balance between the performance and the computation costs.

## 4.4 ABLATION STUDY

To better understand the effect of each component on the output detection result in our method, we analyzed the performances under the following four training models: 1) MTGAN without MCM module; 2) MTGAN without twin-least-square loss module; 3) MTGAN without anomaly rejection loss; 4) MTGAN. In Table 3, we have the following observations.

| Configuration | AUC of $(P_d, P_f)$ | AUC of $(P_f, \tau)$ |
|---|---|---|
| without MCM | 0.99534 | 0.00572 |
| without twin-least-square loss | 0.99544 | 0.00554 |
| without anomaly rejection loss | 0.99654 | 0.00525 |
| MTGAN | **0.99809** | **0.00518** |

Table 3: Average AUC scores of $(P_d, P_f)$ and $(P_f, \tau)$ for component analysis.

The MTGAN achieves higher AUC score of $(P_d, P_f)$ and lower AUC score of $(P_f, \tau)$ than other models. The AUC score of $(P_d, P_f)$ on average compared to other configurations are improved by about 0.276%, 0.266%, and 0.156%, respectively. And the AUC score of $(P_f, \tau)$ on average are optimized by about 10.425%, 6.950%, and 1.351%, respectively. The results indicate the effective-

ness and necessity of MCM, twin-least-square loss, and anomaly rejection loss, which contribute effectively to improve the detection performance.

### 4.5 VISUALIZING THE DETECTION MAPS

For qualitative comparison, as shown in Figure 4, the PAB-DC method almost visually loses all anomalies for the HYDICE data set. Most anomalies are visually mixed with the background in the detection results of AED and LSDM-MoG methods. The RX method can detect almost all anomalies with high intensity, but it cannot suppress background interference well. For the ABU-urban1 data set, the MTGAN can well detect the anomalies with the highest intensity and relatively low false alarm rate in urban scenes. The RX, AAE, ADLR, and AED methods can identify the anomalies locations, but cannot retain the shape of them. For ABU-urban2 data set, both AAE and MTGAN can produce excellent detection results with good background suppression visually, but the ADLR gets poor performance. For the Grand Island data set, the RX, LSDM-MoG, and PAB-DC methods can almost completely detect anomalies but they cannot effectively suppress background. For the EI Segundo data set, MTGAN also shows the best detection performance in both quantitative evaluation and visual analysis.

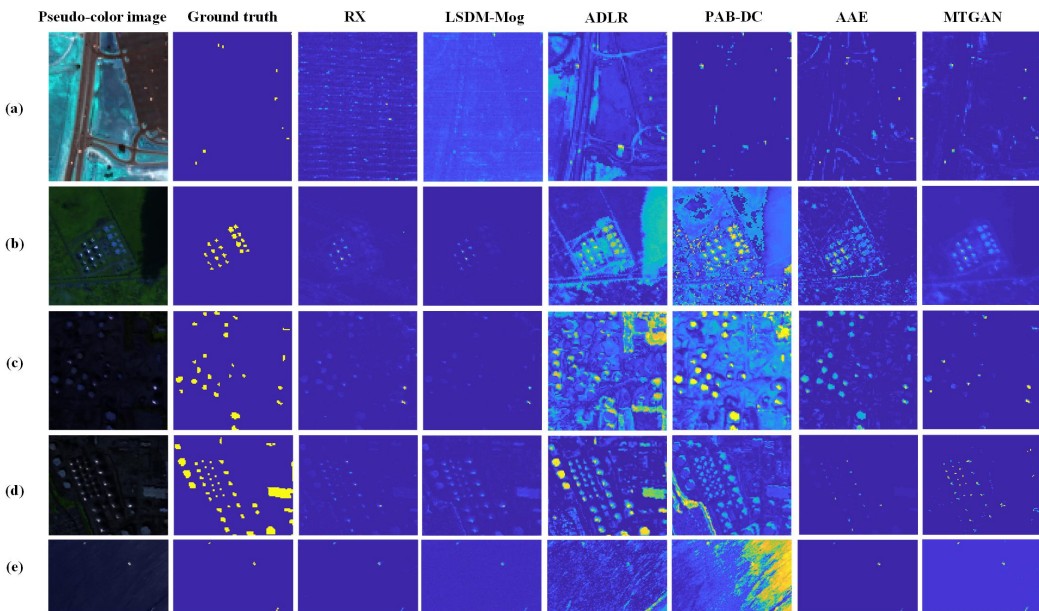

Figure 4: Pseudo-color image, ground truth, and detection maps of MTGAN and the compared methods for (a) HYDICE, (b) ABU-urban1, (c) ABU-urban2, (d) Grand Island, (e) EI Segundo.

## 5 CONCLUSION

In this paper, we propose a new MCM-aware twin-least-squares GAN model for hyperspectral anomaly detection. Several aspects of our framework deserve consideration. For the first time, we propose the MCM strategy to construct multi-scale prior information, reuse and embed the covariance map at multi-scale to create reliable and stable priors adaptively. Therefore, we can solve the lack of priors with pseudo-labeling and make full use of spectral and spatial information. To overcome the problem of gradient vanishing and generate high-quality images, we introduce twin-least-square loss to the architecture in both feature and image domains. Finally, the network enforced with a novel anomaly rejection loss establishes a pure and discriminative background estimation, separating background and anomalies to a greater extent. Through experiments, we have proved that the MTGAN framework exhibits superior performance in background reconstruction and outperforms the state-of-the-art methods.

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
