# OpenReview forum: "MCM-aware Twin-least-square GAN for Hyperspectral Anomaly Detection"
_ICLR.cc/2021/Conference — Reject_

### Official Review · AnonReviewer3 · 2020-10-26
**Ok but not good enough - rejection**

**Rating:** 4
**Confidence:** 4

**Review:**

This paper proposes a MCM-aware twin-least-squares GAN (MTGAN) model for hyperspectral anomaly detection. MTGAN introduce the MCM-aware strategy to construct multi-scale priors with precise second-order statistics, impose the twin-least-square loss on GAN and enforce a new anomaly rejection loss to establish a pure and discriminative background estimation.
Some major comments are as follows:
1.	In the MCM for background dictionary construction component, after generating covariance matrixes, how to obtain the background dictionary? Please state clearly how to construct the background dictionary. How to construct the background dictionary robustly which directly influence the following detection accuracy.
2.	In the twin-least-square for background reconstruction component, the authors impose least square loss instead of cross entropy loss on the two discriminators. As we know, the GAN model usually use the cross entropy loss. It’s not innovation by using least square loss strategy instead of cross entropy loss.
3.	In the experiment part, to make the proposed method MTGAN more convinced, the authors are suggested that using GAN-based anomaly detection methods and background dictionary construction based method as the compared methods.
4.	It is suggested to state clearly in the whole framework, the components of framework should describe more concretely. In fact, we can use traditional methods to obtain the detection results after background dictionary construction. Why using the following GAN-based model to detect and what’s the advantage by using GAN-based method rather than traditional process.
5.	Overall, the MTGAN framework is lack of innovation. The component of MTGAN is independent. We can use the other background dictionary construction methods instead of MCM-aware strategy. Thus, how to design an end-to-end network architecture to solve hyperspectral anomaly detection problem sounds more interesting and novelty.

---

### Official Review · AnonReviewer1 · 2020-10-29
**Accept**

**Rating:** 5
**Confidence:** 4

**Review:**

In this work, the authors find that some emerging methods based on generative adversarial network suffer from the problems of gradient vanishing and training instability with struggling to strike a balance between performance and training sample limitations. To solve this problem, they propose a novel method MTGAN, which introduces the MCM-aware strategy to construct multi-scale priors with precise second-order statistics, and imposes the twin-least-square loss on GAN to help improve the generative ability and training stability in feature and image domains, overcoming the gradient vanishing problem and finally establishes a pure and discriminative background estimation with a new anomaly rejection loss. At the same time, the author also used ablation experiments to prove the effectiveness of each part. But I think this paper is rather unclear, and some symbols are used without explanation, and the explanation of many formulas is not clear enough.

Overall, I vote for 5: Marginally below acceptance threshold. The method is novel and the experiments prove the effectiveness of the method. My major concern is about the clarity of the paper, and the explanation of some formulas is not clear enough.

Pros:
1.The MCM-aware strategy is proposed to reliably and adaptively generate prior dictionaries, which can solve the problem of insufficient samples.
2.The twin-least-square loss is introduced into both the feature and image domains to overcome the gradient vanishing problem.

Cons:
1.This paper is rather unclear, and some symbols are used without explanation, and the explanation of many formulas is not clear enough.
2.Although experiments have proved the effectiveness of the method, it seems that the author rarely explains the reasons

---

### Official Review · AnonReviewer4 · 2020-10-29
**MCM-aware Twin-least-square GAN for Hyperspectral Anomaly Detection**

**Rating:** 5
**Confidence:** 4

**Review:**

[Summary]

In this paper, the authors proposed the MTGAN framework, a GAN-based approach to the task of anomaly detection in hyperspectral images. The main idea behind this work is to exploit twin-least-square loss to perform background modeling in feature and image domains to alleviate the gradient vanishing problem of the previous GAN-based anomaly detection methods. Specifically, they proposed i) an MCM-aware strategy to construct the multi-scale priors, ii) a twin-least-square loss on GAN for training stabilization, and iii) an anomaly rejection loss for background estimation. The experiments on multiple benchmarks show the superiority of the MTGAN to the state of the art hyperspectral anomaly detection methods.

[Strengths]
+ The interesting idea of generating prior dictionaries to alleviate the lack of data for anomaly detection
+ Tackling the GAN's gradient vanishing problem with the proposed twin-least-square loss sounds general-purpose and applicable to other applications
+ Experiments are performed on multiple datasets

[Weaknesses]
- Contributions are not clearly and accurately stated (e.g., the method section is very hard to follow)
- There is a lack of motivation and discussion as well as theoretical analysis on the proposed method
- I am not very familiar with the task of anomaly detection in HSI, but the experimental results seem just marginally better than the other methods
- The English could be improved

[Detailed Questions/Comments]
- More in-depth discussion of the method is necessary, For example: Why does it work? When does it fail?
- Theoretical discussion is missing: e.g., there is no theoretically evidence is provided to support why twin-least-square loss stabilizes the GAN for training compared to the normal discriminator in GAN, etc.
- A discussion on the computational costs (training time, convergence speed) and complexity of the proposed method could be helpful.
- The authors observed that their proposed twin-least-square loss can alleviate the GAN's gradient vanishing problem: it would be nice to evaluate this phenomenon for the generation with GANs and inspect the quality of the generated images
- Providing experimental results on the outlier detection datasets would be very helpful. (see [M1] or an extensive outlier detection benchmark)
- Some important references on the adversarial-discriminator based methods for anomaly/novelty/abnormality/outlier detection are missing: e.g., [M2, M3]

[Missing References]

[M1] On the Evaluation of Unsupervised Outlier Detection: Measures, Datasets, and an Empirical Study by G. O. Campos, A. Zimek, J. Sander, R. J. G. B. Campello, B. Micenková, E. Schubert, I. Assent and M. E. Houle Data Mining and Knowledge Discovery 30(4): 891-927, 2016, DOI: 10.1007/s10618-015-0444-8

[M2] Sabokrou, M., Khalooei, M., Fathy, M., Adeli, E.: Adversarially learned one-class classifier for novelty detection. CVPR(2018)

[M3] Ravanbakhsh, M., et al.: Training adversarial discriminators for cross-channel abnormal event detection in crowds. arXiv preprint arXiv:1706.07680 (2017)

---

### Decision · Program_Chairs · 2021-01-07
**Final Decision**

**Decision:**

Reject

**Comment:**

In this paper, the authors propose a MCM-aware twin-least-squares GAN (MTGAN) model for hyperspectral anomaly detection. The proposed method is somewhat novel, and the efficacy of the proposed method is validated through experiments. However, the clarity of the paper is low, and the explanation of some formulas is not clear enough. Therefore, the quality of the current version is below the acceptance threshold.  I encourage authors to update the paper based on the reviewer's comments and resubmit it to a future venue.